# Curaua–Aramid Hybrid Laminated Composites for Impact Applications: Flexural, Charpy Impact and Elastic Properties

**DOI:** 10.3390/polym14183749

**Published:** 2022-09-08

**Authors:** Natalin Michele Meliande, Michelle Souza Oliveira, Pedro Henrique Poubel Mendonça da Silveira, Rafael Rodrigues Dias, Rubens Lincoln Santana Blazutti Marçal, Sergio Neves Monteiro, Lucio Fabio Cassiano Nascimento

**Affiliations:** 1Department of Materials Science, Military Institute of Engineering-IME, Praça General Tibúrcio, 80, Urca, Rio de Janeiro 22290-270, Brazil; 2Modeling, Metrology, Simulation and Additive Manufacture Section, Brazilian Army Technology Center-CTEx, Avenida das Américas, 28.705, Guaratiba, Rio de Janeiro 23020-470, Brazil; 3Materials Laboratory, Brazilian Army Technology Center-CTEx, Avenida das Américas, 28.705, Guaratiba, Rio de Janeiro 23020-470, Brazil; 4Materials Laboratory, Navy Research Institute-IPqM, Rua Ipiru, nº 02. Cacuia, Ilha do Governador, Rio de Janeiro 21931-090, Brazil

**Keywords:** natural fiber, aramid, curaua non-woven mat, hybrid composite, ballistic helmet, flexural test, Charpy impact test, impulse excitation technique, longitudinal wave velocity, shear wave velocity

## Abstract

Curaua, as a leaf-based natural fiber, appears to be a promising component with aramid fabric reinforcement of hybrid composites. This work deals with the investigation of flexural, impact and elastic properties of non-woven curaua–aramid fabric hybrid epoxy composites. Five configurations of hybrid composites in a curaua non-woven mat with an increasing quantity of layers, up to four layers, were laminated through the conventional hand lay-up method. The proposed configurations were idealized with at least 60 wt% reinforcement in the non-alternating configuration. As a result, it was observed that the flexural strength decreased by 33% and the flexural modulus by 56%. In addition, the energy absorbed in the Charpy impact also decreased in the same proportion as the replaced amount of aramid. Through the impulse excitation technique, it was possible observe that the replacement of the aramid layers with the curaua layers resulted in decreased elastic properties. However, reduction maps revealed proportional advantages in hybridizing the curaua with the aramid fiber. Moreover, the hybrid composite produced an almost continuous and homogeneous material, reducing the possibility of delamination and transverse deformation, which revealed an impact-resistant performance.

## 1. Introduction

The development of ballistic helmet composites with acceptable properties and a minimal environmental impact is a trending research area that could promote sustainable development [1,2]. In this framework, although the use of materials based on natural fibers is an ancient practice, researchers have recently demonstrated renewed interested in this class of material [3,4]. The economic and environmental benefits, associated with their great specific mechanical properties, make natural fiber a recognized candidate for sustainable technological development, especially in the rural areas of developing countries [5,6]. Therefore, researchers are studying the possibility of replacing, partially or even totally, synthetic fibers such glass, carbon or aramid with natural fibers [7,8,9,10].

Laminates with synthetic fibers commonly show significant improvements when joined with different materials. Indeed, hybrid composites with synthetic fiber conjugating with metallic [11] and ceramic [12] materials have been investigated. More important, distinct synthetic fibers are currently being considered for hybrid ballistic composites. Bao et al. [13] performed flexural and ballistic tests on hybrid epoxy composites reinforced with carbon and aramid woven laminates. The highest flexural strength and modulus were found for the carbon-only composite, but a better ballistic performance was found for 30 wt% aramid. In particular, the authors observed a reduction in the flexural strength (21.6%) and modulus (19.1%) for the 50 wt% aramid composite when compared to the carbon-only (0 wt% aramid) composite. It can be inferred from Bao et al.’s [13] work that in hybrid fabric composites, the ballistic performance might not necessarily be tied to mechanical property results. Stephen et al. [14] studied both non-hybrid and hybrid composite laminates with different fabric stacking architecture. The authors observed that the carbon and aramid woven epoxy laminates presented the highest and lowest flexural strength, respectively. The flexural strength of hybrid composites consisting of aramid, carbon and glass was superior to that of only aramid. On the other hand, the aramid-only laminate had the highest energy absorption of all composites. The two-fabric hybrid aramid–glass–aramid composite had about 5% lower energy absorption than the aramid-only composite and was therefore considered ideal for ballistic applications as it offered a 21% reduction in material cost. Yahaya et al. [15] observed that woven kenaf structure produced higher-flexural-strength composites compared to unidirectional and kenaf non-woven mat. Kenaf–aramid hybrid laminated composites showed a predominance of fiber yarn failure along the fiber loading and peeling direction. Despite the pros of this material, such as its low production cost, renewable and abundant sources, biodegradability, CO${}_{2}$-free nature and its position as an income source for many regions in development, natural fibers also have shortcomings and are still considered challenging materials. This is due to the huge variability in their mechanical properties in the same species [16,17]. In this context, the hybridization of synthetic fibers with natural ones in composites can provide a balance between low production cost/environmental impact and better mechanical properties. As such, the disadvantages of one type could be overcome by the advantages of the other. As a result, more sustainable materials with a reasonable production cost and acceptable mechanical properties can be obtained [4,7,18,19,20,21,22,23,24,25,26,27,28,29,30,31,32,33,34,35,36].

One of the natural fibers with recognized outstanding potential is curaua (*Ananas erectifolius)*, which is an Amazonian native plant whose fibers have been used by the local indigenous people in manufactured artifacts. Fibers extracted from the plant leaves are very strong, light and cheap. The literature reports its uses in automotive parts [37], civil construction [38,39] and multilayered ballistic armor system components [40,41]. Table 1 summarizes the main properties of curaua fibers.

Among the strongest synthetic fibers stands the poly (p-Phenylene terephthalate) better known as aramid. The importance of aramid fibers in ballistic applications is well-recognized, owing to their high stiffness and energy absorption capacity and light weight, which are associated with an outstanding ballistic performance [49,50]. The combination of the aforementioned materials and epoxy, which is a matrix suitable for various industrial applications due to its versatility and diversity [51], also allows for the development of an advanced hybrid material for ballistic application. Hence, a previous study aimed to understand the material’s behavior, in relation to its strength and stiffness, when subjected to a transverse load applied in the compression molding direction. The Charpy impact test is part of a preliminary study on the composites’ absorbed energy during impact fracture [52]. Despite the moderate strain rate of this test in contrast to the very high strain rates of the ballistics events, the Charpy test provides some relevant information about the composite failure modes under impact loads and its consequent ability to dissipate energy. In addition to the mechanical properties, obtaining a ballistic helmet and the ballistic and dynamic characteristics is fundamental. Ballistic dynamic deformations occur in the form of stress impulses inside the material associated with elastic, plastic and shock waves. In particular, in elastic waves, the longitudinal and transversal might be considered as main deformation features [53]. One method widely used to investigate elastic properties is the impulse excitation technique (IET), which was once adopted for the application of the simplified non-destructive tests [54].

In a previous study [52], the potential of the hybrid curaua–aramid laminate as a possible ballistic material was preliminary determined. However, a more comprehensive study on the mechanical properties of these materials is necessary. Therefore, the present work investigated the flexural strength and the flexural modulus of hybrid composites, as well as the energy absorbed after the Charpy impact test and a macroscopic analysis of the fracture surface together with reduction maps, revealing the proportional gain obtained by substituting curaua non-woven mat for aramid woven fabric. Moreover, the elastic and shear modulus, as well as Poisson’s ratio and the longitudinal and shear wave velocities of the ballistic laminated composites were determined by the IET.

## 2. Materials and Methods

### 2.1. Materials

The hybrid composite investigated in the present study was made of two types of fiber-based materials: aramid woven fabric, as the synthetic one, and curaua non-woven mat, as the natural one. The aramid woven fabric (Twaron^®^ T750) was produced by Teijin Aramid (Arnhem, The Netherlands) and has been used in ballistic helmets [55]. The as-received aramid fabric was found to have 3360 dtex linear density yarns and an areal density of 460 g/m${}^{2}$. Curaua non-woven mat was supplied by the Brazilian Company Pematec Triangel (Santarém, Brazil) and found to have an areal density of 26 g/m${}^{2}$ and a volumetric density of 0.996 g/cm${}^{3}$.

The polymer matrix used was commercial epoxy of the diglycidyl ether type of bisphenol A (DGEBA) hardened with triethylenetetramine (TETA), in the proportion of 100:13. Both DGEBA and TETA were supplied by Epoxyfiber (Rio de Janeiro, Brazil). The choice of epoxy was due to its low cost, combined with the ease of storage and processing, unlike in conventional ballistic helmets, which use phenolic PVB [49,51].

### 2.2. Composite Processing

In the present work, considering the production of the US personal armor system for ground troop (PASGT) ballistic helmets, currently used by the Brazilian Army, the objective was to reduce costs, total weight and environmental impact associated with the material, without comprimising the ballistic performance. PASGT ballistic helmets are manufactured using a PVB–phenolic matrix, with 19 layers of aramid fabric for reinforcement. Commonly, the composite has a final thickness between 8 and 10 mm and an area density between 10 and 12 kg/m${}^{2}$ [56].

To reduce the number of aramid layers and replace them with a curaua non-woven mat, the configuration adopted in a previous work [52] was used, in which four layers of curaua were used for the complete replacement. In this case, the composite thickness was 9.98 mm, and the areal density was 10.69 kg/m${}^{2}$, keeping within the aforementioned standard for PASGT helmets. Based on this, hybrid configurations of the composites were obtained, in which reinforcements of curaua non-woven mat were added in quantities of 1, 2 and 3 layers, along with 5, 10 and 15 layers of aramid woven fabric, respectively, as shown in Table 2.

All composites were manufactured by the hand lay-up method. Figure 1 illustrates the preparation process for the hybrid composites. Aramid woven fabric and curaua non-woven mat were cut into dimensions of 150 × 120 mm. Prior to processing of the composites, curaua non-woven mat was dried at 70 ${}^{\circ }$C for 24 h for moisture removal. Composite production was carried out using a steel mold with dimensions of 150 × 120 × 11.9 mm. Both the curaua non-woven mat and aramid woven fabric were inserted into the mold with the appropriate number of layers in a non-alternating configuration. In sequence, DGEBA/TETA epoxy, after mixing at a ratio of 100/13, was poured as a still fluid into the mold, between layers. The precise amount corresponding to the desired volume fractions was calculated from the densities of the resin, aramid and curaua. After closing the mold, the composite plate underwent a cold-curing process under a 5-ton load applied by a SKAY hydraulic press (São Paulo, Brazil). Table 3 presents the characteristics of the composites produced and a comparison with the PASGT ballistic helmet.

### 2.3. Flexural Tests

The three-point flexural test was conducted in an INSTRON 5969 universal machine according to ASTM D790-17 Procedure A [57], with a 3 mm/min constant speed and up to 5% maximum strain. A rectangular sample of about 150 × 13 mm and a thickness according to Table 3 was horizontally positioned on two supports with a span length/thickness ratio of about 14. Eight samples were used to perform the test and obtain the stress vs. strain curves, from which the flexural strength (σ_f_) and flexural modulus (E_f_) of the hybrid composites were determined.

### 2.4. Charpy Impact Test

To investigate the impact strength and the energy absorption, a series of Charpy impact tests were performed, followed by macroscopic characterization of the damage associated with the fractured sample. The tests were performed in a hammer pendulum in which the impact load was applied along the thickness of the specimen. The output of the test was the impact strength, or the impact toughness, defined as the threshold of force per unit area before the material underwent a fracture. As impact energy reached a threshold value that the specimen could accommodate, the material underwent a failure involving fracture damages. The Charpy impact was performed in a Tinius Olsen Impact 104 machine, in accordance with ASTM D6110–18 [58]. Ten specimens with dimensions of 120 × 12.7 mm and thicknesses according to Table 3 were used for each group.

### 2.5. Impulse Excitation Technique

One of the ways to obtain the propagation velocity of the elastic wave, without performing the dynamic test, is through the analysis of elastic moduli, as obtained in the Sonelastic ^®^ (Ribeirao Preto, São Paulo, Brazil) machine. The technique consists of impulse excitation. The elastic modules and damping are characterized by the acoustic response that the sample emits after receiving a slight mechanical impulse. This response contains the natural frequencies of vibration, which are proportional to the modulus of elasticity, and shows a rate of attenuation that is similar to damping [59]. In order to determine the elastic modulus of the hybrid laminated composites, the impulse excitation test was performed on the samples. The test was carried out according to ASTM E-1876-09 [60], in which one sample from each configuration was used, and ten tests were performed on each sample.

The samples were produced with dimensions of 113 × 20 mm and thicknesses according to Table 3. The following equations were used to determine the Young’s modulus (*E*), using shear modulus (*G*), Poisson’s coefficient (μ), longitudinal sound velocity (*V_p_*) and sound velocity of the shear modulus (*V_s_*):(1)$$E=0.9465\cdot \left(\frac{mf_{f}^{2}}{b}\right)\cdot \left(\frac{L^{3}}{t^{3}}\right)\cdot T_{1}$$
(2)$$G=\frac{4Lmf_{t}^{2}}{bt}\cdot R$$
(3)$$\mu =\frac{E}{2G}-1$$
(4)$$V_{p}=\sqrt{\frac{4G-E}{\rho \cdot (3-\frac{E}{G})}}$$
(5)$$V_{s}=\sqrt{\frac{G}{\rho }}$$
where *m* is the mass of the bar, *b* is the width of the bar, *L* is the length of the bar, *t* is the thickness of the bar, *f_t_* is the fundamental frequency of the bar in torsional mode, and *R* and *T*_1_ are the correction factors.

## 3. Results and Discussion

### 3.1. Flexural Results

Flexural stress is normally the result of the action of transverse loads that tend to bend the material and generate an approximately linear stress distribution within it [61]. This distribution alternates between tensile and compressive stresses in the same cross section, considering a homogeneous material [62,63]. In the case the specimen is made up of different materials, with different moduli of elasticity, the stress distribution also becomes different. Hence, it justifies the importance of studying the flexural properties of hybrid composites. Figure 2 shows the flexural behavior of all hybrid composites proposed through the stress vs. strain curve. Interestingly, as seen for the tensile properties [52], all the composites followed a similar trend for the flexural strength.

As can be seen in Figure 2 by the decays of the curves before the plateau, all composites started to experience a decrease in flexural strength before the 5% strain limit, which is accepted by ASTM D790-17 [57]. It is well-known that when a composite begins to fail, its ability to resist the increase in deflection is reduced. From this point on, an attempt to internally reorganize the material occurs, with a redistribution of stresses, where a series of increases in strain occurs without an increase in stress. Depending on the extent of this failure, additional strains may no longer be accommodated by the material, causing the failure of the composite and the consequently reduction in stress. This process was observed for all hybrid composites and both E-0A/4C and E-19A/0C. The literature reports that this stress reduction is directly associated with fiber delamination and fiber deformation [62].

Composite failure is attenuated as the number of aramid woven fabric layers is reduced and the number of curaua non-woven mat layers is increased [52]. This could be explained by the reduction in available interfaces for delamination and the deformation capacity, in addition to the increase in the dominance of the non-woven mat structure. The curaua non-woven mat has voids and discontinuities through which the resin can penetrate, allowing for better wettability. This, associated with the random distribution of the fibers, produces a continuous and homogeneous material, which reduces delamination [64,65]. All this contributes to explaining the increase in flexural strength with increasing aramid layers, as shown in Figure 3. On the other hand, the increase in thickness causes an increase in the flexural strength. These two opposing effects are clearly shown in Figure 2, in which tests were interrupted at 5% flexural strain. However, substitution of the curaua mat for aramid fabric was found to substantially increase the flexural strain.

Figure 3 and Table 4 show the comparison of the results for flexural strength and flexural modulus.

A hybrid composite features unique characteristics that can be used to meet different requirements for different applications with regard to its mechanical properties. A key parameter in the structure of the hybrid composite is the arrangement of the hybrid reinforcements inside the matrix. Factors such as hybrid reinforcement arrangement, number of layers and reinforcement content directly affect various properties, such as the strength and flexural modulus, as well as fatigue behavior and the impact resistance of the hybrid composite [63,66].

Zachariah et al. [67] pointed out the influence of the hybrid fiber setup on the laminate response to the static traverse loading. For the composite E-0A/4C, replacing one layer of curaua non-woven mat with five layers of aramid woven fabric (E-5A/3C) resulted in a decrease of 36% in the σ_f_. The increase in available interfaces for delamination, added to the deformation capacity due to the replacement of curaua layers by aramid, did not compensate for the reduction in composite thickness. Delamination and deformation did not occur with enough intensity, as observed in composites with higher number of aramid layers, precisely because of the greater thickness of the composite, which impairs deflection. Consequently, these two mechanisms considerably reduced the flexural strength, as shown in Figure 3 and Table 4. Considering the replacement of one layer of curaua for every five layers of aramid (E-10A/2C), the reduction in the value of *σ_f_* was less significant. Delamination and deformation of the aramid yarns began to take place more effectively, compensating for the observed thickness reduction, and thus increasing the material’s flexural strength.

Replacing the curaua with aramid layers caused an increase in the flexural strength of the E-15A/1C and E-19A/0C composites. However, it is important to note that from the composite E-10A/2C to the composite E-15A/1C, this increase was not significant, unlike the increase observed from the composite E-15A/1C to the composite E-19A/0C. In order to corroborate these observations and verify whether there were significant differences in the composites’ flexural strength, statistical analysis was carried out via analysis of variance (ANOVA) at a 5% significance level. The results are shown in Table 5.

The ANOVA results of F >> F_critic_, shown in Table 5, show with 95% confidence that the composites had different *σ_f_*. In order to compare the composites’ *σ_f_* to each other, Tukey’s test was applied based on the minimum significant difference (MSD) that must exist between two averages. The differences between the *σ_f_* of E-15A/1C, E-10A/2C and E-0A/4C, considering the pairs, were smaller than the MSD. The differences between the other unmentioned groups were greater than the MSD value, supporting the hypothesis that the *σ_f_* of composites E-15A/1C, E-10A/2C and E-0A/4C can be considered statistically similar.

Based on Figure 3 and Table 4, it can also be seen that the flexural modulus decreased with the reduction in the number of aramid layers and the increase in the number of curaua layers. It should be noted that there was no significant variation in *E_f_* between E-10A/2C and E-0A/4C. In order to corroborate these observations, ANOVA was also performed with a significance level of 5%, and the results are shown in Table 6. It is possible to state with 95% confidence that the *E_f_* averages were statistically different, once F >> F_critic_. Using Tukey’s test, it was found that only the difference between E-10A/2C and E-0A/4C averages was smaller than the MSD, indicating that these averages are statistically equal.

### 3.2. Charpy Impact Test and Damage Analysis

Impact strength tests are commonly used to evaluate the amount of energy absorbed by materials before fracturing using the Charpy or Izod configurations [4,67,68,69,70]. In previous Charpy impact results [52], the energy absorbed was inversely proportional to the number of curaua non-woven mat layers. A comparison between the impact and the flexural behavior of composite samples with different volume reductions in the aramid layers in terms of flexural and impact strength values is depicted in Figure 4. This comparison is significant as it gives an idea of the trade-off between the static and the dynamic transverse behavior when adding hybrid fibers to the composite laminate. Based on this figure, the absorbed energy decreased by about 23% with the insertion of one layer of curaua non-woven mat, 49% for two layers, 69% for three layers and 88% for total aramid fabric substitution. Intriguingly, it can be observed that the energy absorbed decreased in a similar proportion as the aramid amount reduction (Table 4), indicating that this property is controlled by the aramid fabric and is independent of the curaua non-woven mat.

Regarding the flexural–impact strength comparison in Figure 4, E-19A/0C exhibited an impact strength of 996 kJ/m${}^{2}$, which is 10 times higher than that of E-0A/4C. On the other hand, its flexural strength was 109 MPa, 37% higher than that of E-0A/4C. Regarding the hybrid composites, E-15A/1C showed a higher flexural strength, 72 MPa, and impact strength, 704 kJ/m${}^{2}$. In order to complement the previous results [52], the ANOVA was used with a 5% significance level to verify whether there were significant differences in the composites’ absorbed energies. Table 7 presents the ANOVA and Tukey test for the Charpy impact energy absorbed. As F >> F${}_{\mathrm{critic}}$, it is possible to state with 95% confidence that there were statistically different averages. In order to compare the composites’ absorbed energies, the Tukey test was used based on the MSD with a 5% significance level. All the composites’ energy average differences were greater than the MSD, certifying that the decrease in energy absorbed was proportional to the reduction in aramid layers and inversely proportional to the increase in curaua layers. This result was already expected based on aramid’s superior mechanical properties compared to natural fibers, especially regarding the impact strength [71].

The failure mode analysis after the Charpy impact tests was quite revealing and explains in detail the difference in impact strength, which translates into the failure mechanisms of the composites. Figure 5 shows the fracture modes of the investigated laminates.

It can be seen that the E-19A/0C laminate had intense interlaminar debonding, also known as delamination, which involved several layers. This could be explained by the fact that the energy needed to separate the aramid layers was significantly lower than the energy needed to break the fabric yarns by tensile and bending or even shear. Thus, at the beginning of the impact, considerable energy was consumed in the process of complete delamination of the composite [72]. In addition, some fabric yarns showed the mechanisms of peel-up delamination, i.e., they were also broken or separated in this process. However, in macro terms, as seen in Figure 5, delamination was clearly the dominant failure mode, with debonding as its initiating mechanism. The complexity of the fracture can be observed by taking into account the inset of Figure 5a. On the y axis, the layers are axially separated in relation to the laminate axis, while in the x axis, these fibers are pulled, causing the laminate to bend. Laminate E-15A/1C exhibited the same behavior for aramid layers, but the transverse interlaminar crack was also evident. Transverse cracks in laminate usually occur at a stress which is much lower than the laminate strength. As the laminate stress increases, the number of transverse cracks increases [73]. This is due to the preferential fracture that occurs in the curaua non-woven mat. As for the E-10A/2C laminate, the fracture mechanism called translaminar crack was observed. The E-5A/3C laminate stands out due to the evidence it exhibits for the longitudinal interlaminar crack mechanism. Intralaminar damage leads to delamination crack branching, which pulls out fiber bridging and dissipates more energy during delamination propagation [74]. Finally, E-0A/4C presented the lowest impact energy, as well as the least amount of complex fracturing, and, therefore, is considered less flexible.

### 3.3. Impulse Excitation Technique (IET) Analysis

Table 8 shows the results for Young’s modulus (E), shear modulus (G), Poisson’s ratio (ν), longitudinal wave velocity (V_p_) and shear wave velocity (V_s_) of the hybrid composites. Based on this table, one can see that the Young’s modulus decreased with the reduction in the number of aramid layers. It should also be noted that there was no significant reduction in *E* from the E-10A/2C composite to the E-5A/3C composite. In order to corroborate these observations and verify whether there were significant differences between the *E* of the composites, ANOVA was used with a significance level of 5%.

Table 9 shows that F >> F_critical_. It can be stated with 95% confidence that there were statistically different averages. All the differences between the *E* of the composites were greater than the MSD, except for the E-10A/2C and E-5A/3C composites. This confirms the preliminary observation that the Young’s moduli of the E-10A/2C and E-5A/3C composites were similar, despite the lower relative amount of aramid in the E-5A/3C composite.

The shear modulus also decreased with the reduction in the number of aramid layers. Similarly to the Young’s modulus, there was no significant reduction in *G* from the E-10A/2C composite to the E-5A/3C composite. The ANOVA and Tukey test were applied for this parameter as well, both with a significance level of 5%, as presented in Table 10. Therefore, as F >> F_critical_, it can be said with 95% confidence that there were statistically different averages. All the differences between the *G* of the composites were greater than the MSD, except for the composites E-10A/2C and E-5A/3C.

Based on Table 8, taking the E-19A/0C composite as a reference and comparing it with the E-15A/1C and E-10A/2C composites, there was a reduction in the amount of aramid of about 29% and 55%, as aforementioned, while the shear modulus decreased by about 32% for the first and 53% for the second. On the other hand, from the E-19A/0C composite to the E-5A/3C composite, there was a reduction of about 79% in the amount of aramid, while the shear modulus decreased by only about 53%. Considering the E-0A/4C composite, the introduction of five layers of aramid (E-5A/3C composite) significantly increased the shear modulus by about 122%. The introduction of five more layers of aramid (E-10A/2C composite) did not significantly change the value of *G*, despite the increase in the amount of aramid. It is important to highlight that the *G* values of the E-19A/0C composite, obtained by the IET, and those obtained for a similar composite based on epoxy–aramid [75] had a comparable order of magnitude, i.e., 109 Pa. This reinforces the validity of the test and supports the results obtained for the shear modulus of the other composites.

The *E* and *G*, as well as Poisson’s ratio, also decreased with the introduction of a curaua layer. As shown in Table 11, the F >> F_critical_, i.e., there were statistically different averages. All the differences between the ν of the composites were smaller than the MSD, except when compared with the E-19A/0C composite. This verifies the preliminary observations that there was a reduction in the ν with the introduction of a layer of curaua (E-15A/1C), but the ν of the composites E-15A/1C to E-0A/4C was similar.

Longitudinal and shear wave velocities also decreased with the reduction in the number of aramid layers. One should note that there was no significant reduction in the V_P_ or V_S_ between E-10A/2C and E-5A/3C composites. Table 12 shows that F >> F_critical_, i.e., there were statistically different averages. The difference between the V_P_ of the E-19A/0C and E-15A/1C composites was smaller than the MSD, indicating statistical equality. Likewise, between the V_P_ of the E-15A/1C, E-10A/2C and E-5A/3C composites, no significant variation in V_P_ was found. Nonetheless, the differences between the E-19A/0C composite, as the reference, and the E-10A/2C and E-5A/3C composites were greater than the MSD, which indicates that the observed reductions in the V_P_ of E-19A/0C for E-10A/2C and E-5A/3C composites were significant. Regarding the E-0A/4C composite, there was a significant reduction in V_P_ compared to composites E-19A/0C and E-15A/1C, but the same was not observed for the other composites.

As for E, G, ν and V_P_, Table 13 shows the ANOVA and Tukey test for V_S_. Once the F >> F_critical_, it can be stated with 95% confidence that there are statistically different averages. All the differences between the V_S_ of the composites were greater than the MSD, except for E-10A/2C and E-5A/3C composites. This confirms the preliminary observation that the shear wave velocities of the E-10A/2C and E-5A/3C composites were similar, despite the lower relative amount of aramid in the E-5A/3C composite.

It is suggested that the hybridization of aramid fibers with curaua non-woven mat in epoxy matrix composites was impaired in terms of Young’s and shear moduli values. Although, for E-5A/3C composite, the amount of aramid decreased considerably more than these properties compared to the E-19A/0C composite, the reductions in the Young’s and shear moduli were significant, around the order of 50%. This might greatly affect the system performance in terms of rigidity. In regard to the E-0A/4C composite, the introduction of five layers of aramid (E-5A/3C composite) significantly increased the Young’s and shear moduli by more than 110%. Meanwhile, regarding Poisson’s ratio, comparing the E-19A/0C composite with the E-15A/1C and E-0A/4C composites, there was a reduction in the amount of aramid ranging from about 29% to 79%, while the ν decreased by a maximum of 35%. It is important to note that, from the E-15A/1C composite to the E-0A/4C composite, there were no significant variations in terms of ν. Regarding the longitudinal and shear wave velocities, the E-15A/1C composite presented the greatest performance. This is due to the fact that it had a reduction of 29% in the amount of aramid, when compared with the E-19A/0C composite, while the V_P_ did not vary significantly, and the V_S_ decreased by only about 17%. These properties are important for the ballistic performance of laminated composites as they directly affect their ability to distribute impact energy within the same layer and among adjacent layers and, consequently, the deformation capacity of the material.

### 3.4. Reduction Maps

As proposed elsewhere [52], the concept of reduction maps was employed in this work to analyze the effect of the reduction in aramid layers on the properties of flexural strength, the flexural modulus and the energy absorbed during the Charpy impact test. The reduction maps, illustrated in Figure 6, were constructed by adopting the values of the flexural and impact properties observed in this work as a function of the reduction in the aramid volume of the composites, described in Table 4. It is important to mention that in this work, the negative limit condition that could occur was considered, i.e., the aramid laminates were put in the upper region, which was the region in direct contact with the flexural point load. While the curaua non-woven mats were placed in the tensile region, the lower region was in contact with the supports. Based on a literature report [76], greater flexural properties of a hybrid composite are obtained when the natural fiber is placed in the compression region and the synthetic fiber is placed in the tensile region.

Figure 6a shows the reduction maps for the investigated flexural strength properties. These maps plot the reduction value of the property as a function of the aramid volume in the composite, both for the mean value (red solid square) and the lower limit (red open circle). It can be observed in Figure 6a that composites E-10A/2C, E-5A/3C and E-0A/4C were in the “gain zone”. The gain zone represents a proportionally favorable condition of the analyzed property as a function of the reduction in the aramid volume. Only composite E-15A/1C presented a flexural strength within the “loss zone”, in which the reduction in the mean value of the property of the composite became unfavorable compared to that of the control group (E-19A/0C). On the other hand, the lower limit of composite E-15A/1C was in the “gain zone”. As mentioned earlier, when considering a hybrid composite, the so-called neutral line of the material is shifted due to the influence of the materials, and this characteristic was well-evidenced in the E-15A/1C composite.

As for the flexural modulus, little reduction was observed, being the largest for the E-0A/4C condition in the order of 7%. Intriguingly, a similar behavior was observed for the reduction in impact energy, on the order of 11% for the E-0A/4C composite. In both properties, it was possible to observe that the lower limit, the value of the difference in the average of the analyzed composite plus the standard deviation of the average of the control group minus its standard deviation, was much greater than the mean reduction. As follows, Figure 7 illustrates the reduction maps with the results obtained with the IET.

Figure 7a shows that E-15A/1C and E-10A/2C presented Young’s moduli within the “loss zone”. A reduction in the amount of aramid of about 29% and 55% from the E-19A/0C composite to the E-15A/1C and E-10A/2C composites was observed, while the *E* decreased by about 36% and 57%, respectively. For the E-15A/1C composite, the specific Young’s modulus decreased by about 31%, while for the E-10A/2C composite, this decrease was about 51% [52]. On the other hand, from the E-19A/0C composite to the E-5A/3C composite, there was a reduction of about 79% in the amount of aramid, while the *E* decreased by only about 59%. In specific terms, this reduction was around 50% smaller [52]. Considering the E-0A/4C composite as a reference, the introduction of five layers of aramid (E-5A/3C composite) significantly increased the *E*, by about 112%. Regarding the *G*,as seen in Figure 7, the only laminate composite located in the “loss zone” was E-15A/1C, with a reduction of 32%. Additionally, the ν decreased by a maximum of 35% for E-5A/3C. It is important to highlight that the ν values for the E-19A/0C composite (0.59), obtained in the IET, and those obtained for a similar epoxy–aramid composite (0.69) [75], were similar. The highest reduction seen, taking into account the V_P_, was for the E-0A/4C composite. Meanwhile, the V_S_ decreased by only about 17% after replacing four aramid layers with one layer of curaua non-woven mat. As for E-10A/2C and E-5A/3C composites, compared with E-19A/0C, they showed a reduction in the aramid amount of about 55% and 79%, respectively, while the V_P_ and V_S_ decreased by only about 30%. Considering the E-0A/4C composite, the introduction of five layers of aramid significantly increased the V_P_ and V_S_ by about 40%. The introduction of five more layers of aramid (E-10A/2C composite) was hindering, showing an increase of about 108% in the amount of aramid, while V_P_ and V_S_ did not vary significantly. Thus, more experimental or even simulation ballistic results should be obtained for this material in order to confirm its use in ballistic helmets.

## 4. Conclusions

In this paper, the flexural, impact properties and elastic modulus of hybrid laminated curaua mat and aramid-fabric-reinforced epoxy composites were investigated, aiming at future application in ballistic helmets. The replacement of aramid layers by curaua resulted in a reduction in flexural strength as the number of curaua layers was increased. This downward trend also occurred for the flexural modulus of the hybrid composites. Similar to flexural strength and flexural modulus, the impact strength of the hybrid composites showed a decrease as a result of the reduction in aramid layers. Macroscopic analysis of fractured specimens revealed that the main mechanisms of impact energy absorption in the laminate composite were interlaminar delamination and deformation of the fibers to rupture. The reduced number of aramid layers resulted in a lower occurrence of delamination, causing a lower energy absorption capacity. From the IET results, it was observed that the replacement of the aramid layers by the curaua layers resulted in the reduction in all parameters related to the elastic properties of the hybrid composites, similar to the reduction in the values in the flexural and Charpy impact tests. Reduction maps demonstrated a percentage gain in the elastic moduli of E-5A/3C and E-0A/4C, which indicates the substitution of aramid for curaua layers is advantageous. Regarding the flexural and impact properties of the composites, the reduction maps indicated that the addition of curaua layers evidenced a more favorable situation with respect to this type of applied stress, which might also be advantageous for ballistic helmets.

## Figures and Tables

**Figure 1 polymers-14-03749-f001:**
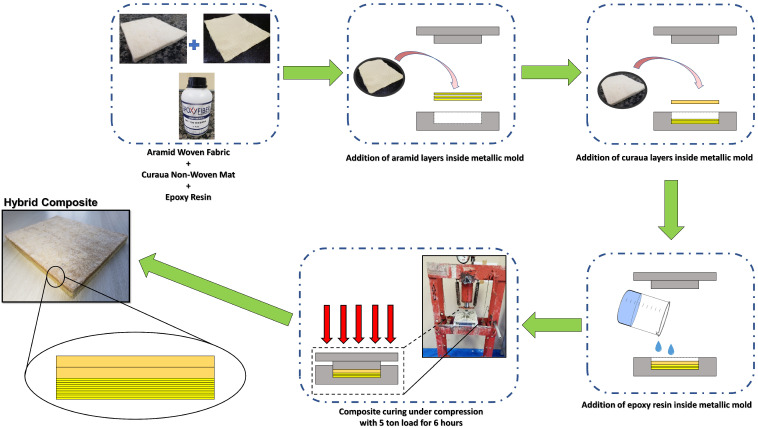
Scheme of production process for hybrid laminated composites.

**Figure 2 polymers-14-03749-f002:**
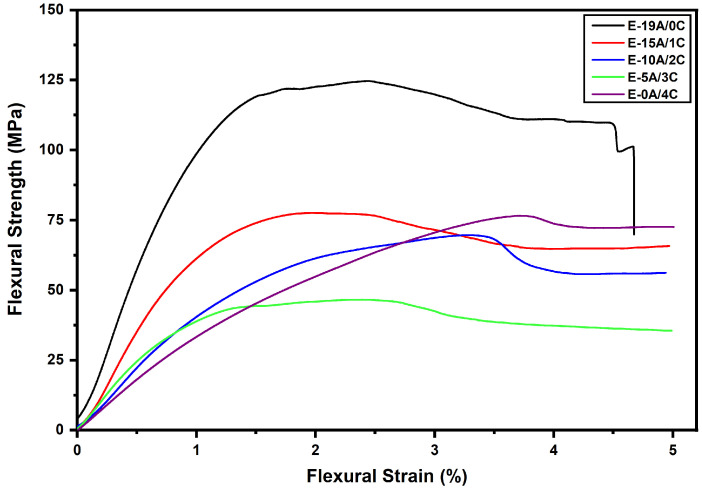
Stress–strain curves of three-point flexural test.

**Figure 3 polymers-14-03749-f003:**
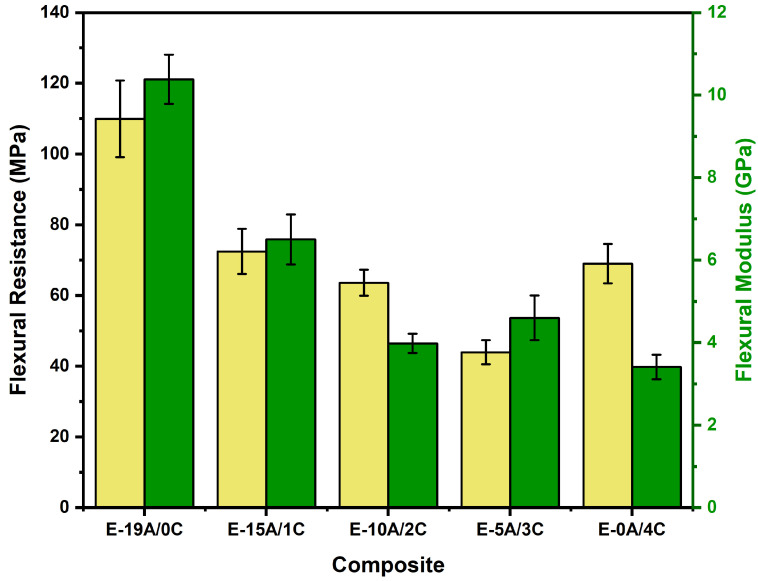
Comparison between flexural strength (*σ_f_*) and the flexural modulus (*E_f_*) values of hybrid composites.

**Figure 4 polymers-14-03749-f004:**
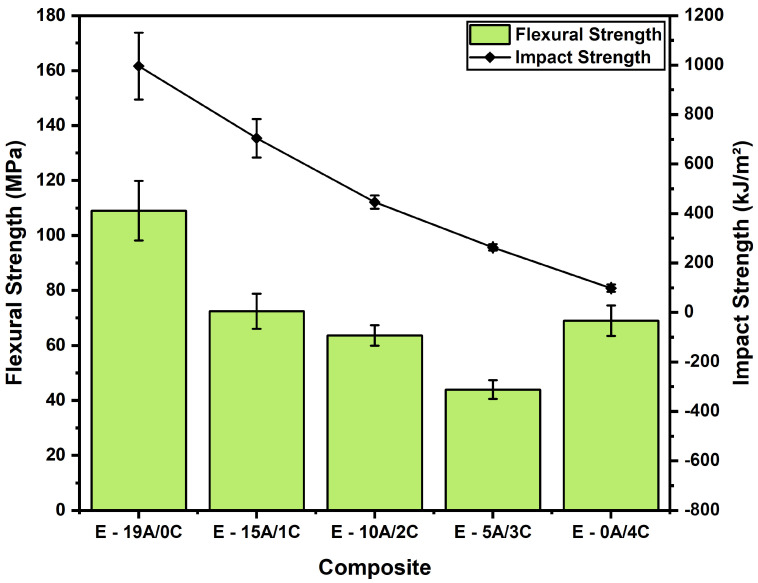
Comparison of flexural and impact strength results of hybrid laminated composites.

**Figure 5 polymers-14-03749-f005:**
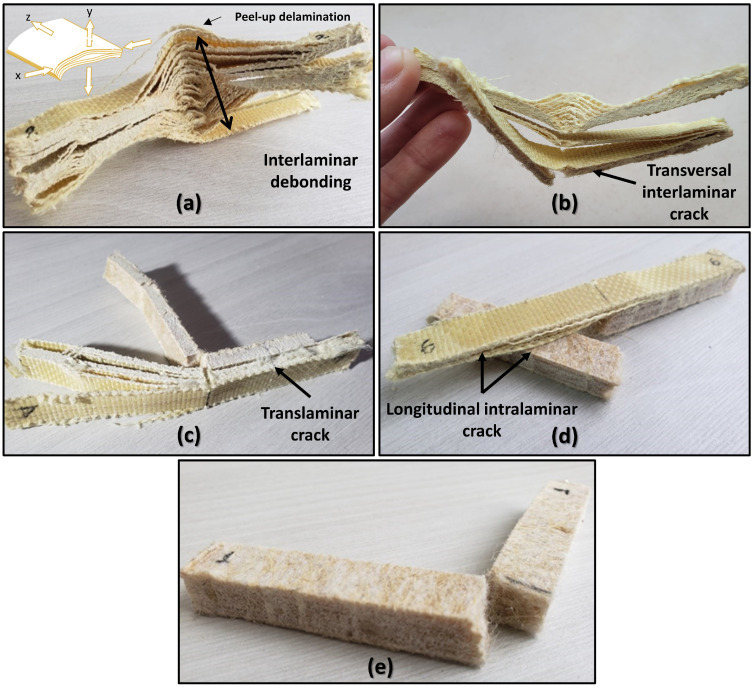
Macroscopic failure mode analysis after the Charpy impact test: (**a**) E-19A/0C; (**b**) E-15A/1C; (**c**) E-10A/2C; (**d**) E-5A/3C; and (**e**) E-0A/4C.

**Figure 6 polymers-14-03749-f006:**
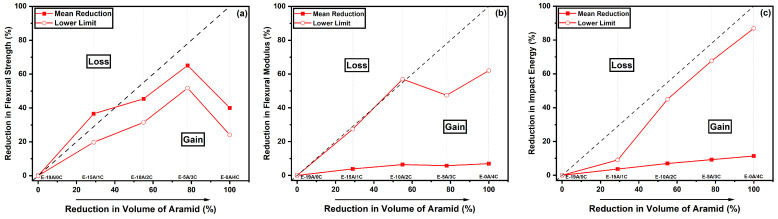
Reduction maps for flexural and impact properties of hybrid curaua–aramid composites: (**a**) flexural strength (σ_f_); (**b**) flexural modulus (E_f_); and (**c**) impact energy.

**Figure 7 polymers-14-03749-f007:**
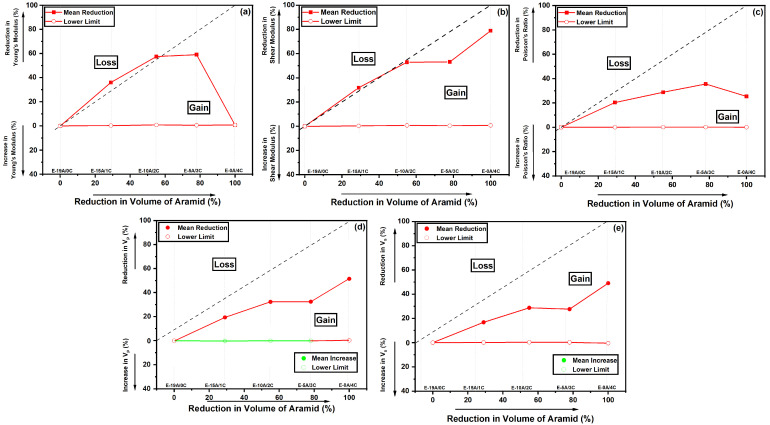
Reduction maps for elastic moduli of hybrid curaua–aramid composites: (**a**) Young’s modulus; (**b**) shear modulus; (**c**) Poisson’s ratio; (**d**) V_p_; and (**e**) V_s_.

**Table 1 polymers-14-03749-t001:** Physical, chemical and mechanical properties of curaua fibers [42,43,44,45,46,47,48].

**Curaua**	**Cellulose** **(wt%)**	**Hemicellulose** **(wt%)**	**Lignin** **(wt%)**	**Wax** **(wt%)**	**Pectin** **(wt%)**	**Ash** **(wt%)**
	73.6	9.9	7.5	-	-	-
	**Microfibril** **Angle (${}^{\circ }$)**	**Density** **(g/cm^3^)**	**Fiber** **Diameter** **(μm)**	**Tensile** **Strength** **(MPa)**	**Specific** **Strength** **(S/ρ)**	**Elongation** **at Break** **(%)**
	15	1.4	170	158–729	113–521	5

**Table 2 polymers-14-03749-t002:** Composite configurations based on the number of reinforcing layers.

Composite Configuration	Number of Layers	
	Aramid Woven Fabric (A)	Curaua Non-Woven Mat (C)
E-19A/0C	19	0
E-15A/1C	15	1
E-10A/2C	10	2
E-5A/3C	5	3
E-0A/4C	0	4

**Table 3 polymers-14-03749-t003:** Composite’s parameters associated with distinct configuration layers.

Composite	Weight (g)	Thickness (mm)	Vol.% Total Reinforcement	Vol.% Aramid	Vol.% Curaua	Areal Density (Kg/m^2^)
PASGT-based	~200	~ 8–10	~70	-	-	11.24
E-19A/0C	197.78 ± 1.41	8.32 ± 0.03	73.29 ± 0.58	73.29 ± 0.58	0	10.99 ± 0.07
E-15A/1C	204.65 ± 0.92	9.06 ± 0.04	68.52 ± 0.63	52.06 ± 0.39	16.47 ± 0.44	11.37 ± 0.51
E-10A/2C	200.47 ± 2.83	9.37 ± 0.08	65.00 ± 0.83	33.27 ± 0.40	31.73 ± 0.76	11.14 ± 0.16
E-5A/3C	196.14 ± 2.17	9.61 ± 0.13	61.22 ± 0.59	15.96 ± 0.15	45.26 ± 0.72	10.90 ± 0.12
E-0A/4C	192.41 ± 2.21	9.98 ± 0.12	57.97 ± 0.86	0	57.97 ± 0.86	10.69 ± 0.12

**Table 4 polymers-14-03749-t004:** Flexural properties of hybrid laminated composites.

Composite	Aramid Volume Reduction (%)	$\sigma_{f}$ (MPa)	E_f_ (GPa)
E-19A/0C	0	109.02 ± 10.83	10.38 ± 0.60
E-15A/1C	29	72.44 ± 6.38	6.50 ± 0.60
E-10A/2C	55	63.60 ± 3.71	3.98 ± 0.23
E-5A/3C	78	43.94 ± 3.42	4.60 ± 0.54
E-0A/4C	100	68.98 ± 5.58	3.41 ± 0.30

**Table 5 polymers-14-03749-t005:** ANOVA and Tukey’s test parameters for composites’ flexural strength.

Mean Treatment Squares	Mean Residue Squares	F (Calculated)	F_critic_ (Tabulated ^1^)	q (Tabulated ^2^)	MSD
4473.80	42.92	104.24	2.65	4.07	9.43

^1^ Snedecor’s F distribution with 5% significance. ^2^ Student’s t distribution with 5% significance.

**Table 6 polymers-14-03749-t006:** ANOVA and Tukey’s test statistical parameters for composites’ flexural modulus.

Mean Treatment Squares	Mean Residue Squares	F (Calculated)	F_critic_ (Tabulated ^1^)	q (Tabulated ^2^)	MSD
63.86	0.231	276.19	2.65	4.07	0.69

^1^ Snedecor’s F distribution with 5% significance. ^2^ Student’s t distribution with 5% significance.

**Table 7 polymers-14-03749-t007:** ANOVA and Tukey test parameters for the Charpy impact tests.

Mean Treatment Squares	Mean Residue Squares	F (Calculated)	F_critic_ (Tabulated ^1^)	q (Tabulated ^2^)	MSD
142,858,001	825,184	173.12	2.69	4.10	1.41

^1^ Snedecor’s F distribution with 5% significance. ^2^ Student’s t distribution with 5% significance.

**Table 8 polymers-14-03749-t008:** Mechanical properties of composites obtained from the impulse excitation test.

Composite	Young’s Modulus E (GPa)	Shear Modulus G (GPa)	Poisson’s Ratio ν	Longitudinal Wave Velocity V_p_ (m/s)	Shear Wave Velocity V_s_ (m/s)
E-19A/0C	17.64 ± 0.54	5.59 ± 0.14	0.59 ± 0.05	4110 ± 960	2111 ± 63
E-15A/1C	11.28 ± 0.36	3.81 ± 0.08	0.47 ± 0.04	3314 ± 774	1756 ± 41
E-10A/2C	7.49 ± 0.25	2.63 ± 0.04	0.42 ± 0.04	2783 ± 652	1504 ± 25
E-5A/3C	7.22 ± 0.35	2.62 ± 0.08	0.38 ± 0.07	2778 ± 660	1527 ± 55
E-0A/4C	3.41 ± 0.22	1.18 ± 0.03	0.44 ± 0.08	1994 ± 332	1075 ± 31

**Table 9 polymers-14-03749-t009:** ANOVA and Tukey’s test parameters for the Young’s modulus.

Mean Treatment Squares	Mean Residue Squares	F (Calculated)	F_critic_ (Tabulated ^1^)	q (Tabulated ^2^)	MSD
289.28	0.13	2208	2.58	4.02	0.46

^1^ Snedecor’s F distribution with 5% significance. ^2^ Student’s t distribution with 5% significance.

**Table 10 polymers-14-03749-t010:** ANOVA and Tukey’s test parameters for the shear modulus.

Mean Treatment Squares	Mean Residue Squares	F (Calculated)	F_critic_ (Tabulated ^1^)	q (Tabulated ^2^)	MSD
27.05	0.01	3875	2.58	4.02	0.11

^1^ Snedecor’s F distribution with 5% significance. ^2^ Student’s t distribution with 5% significance.

**Table 11 polymers-14-03749-t011:** ANOVA and Tukey’s test parameters for Poisson’s ratio.

Mean Treatment Squares	Mean Residue Squares	F (Calculated)	F_critic_ (Tabulated ^1^)	q (Tabulated ^2^)	MSD
0.06	0.01	18.68	2.58	4.02	0.07

^1^ Snedecor’s F distribution with 5% significance. ^2^ Student’s t distribution with 5% significance.

**Table 12 polymers-14-03749-t012:** ANOVA and Tukey’s test parameters for longitudinal wave velocity.

Mean Treatment Squares	Mean Residue Squares	F (Calculated)	F_critic_ (Tabulated ^1^)	q (Tabulated ^2^)	MSD
6,097,543	498,321	12.24	2.58	4.02	899

^1^ Snedecor’s F distribution with 5% significance. ^2^ Student’s t distribution with 5% significance.

**Table 13 polymers-14-03749-t013:** ANOVA and Tukey’s test parameters for shear wave velocity.

Mean Treatment Squares	Mean Residue Squares	F (Calculated)	F_critic_ (Tabulated ^1^)	q (Tabulated ^2^)	MSD
1,438,705	2052	701	2.58	4.02	57.66

^1^ Snedecor’s F distribution with 5% significance. ^2^ Student’s t distribution with 5% significance.

## Data Availability

The data presented in this study are available on request from the corresponding author.

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
