# Peer review of "Curaua–Aramid Hybrid Laminated Composites for Impact Applications: Flexural, Charpy Impact and Elastic Properties"

_polymers, 2022, doi:10.3390/polym14183749_

Round 1
Reviewer 1 Report
This manuscript presents the flexural characterization of hybrid laminated composites made of aramid and curaua natural fibers and epoxy resin. The composites are intended to be used as an armor material for helmet applications. The manuscript reports the methodology to test the composite experimentally. In general, the results are clear, well discussed, and of interest to readers of Polymers; however, the manuscript needs improvement. Therefore, the following comments should be addressed before the manuscript can be recommended for publication:
General comments
-The manuscript has several grammar, style, and syntax errors. The text is hard to read in the abstract and introduction. The manuscript should be checked thoroughly for grammatical mistakes and proofread by a native English speaker before it can be recommended for publication.
Introduction
-The authors should add a few more references regarding hybrid composites for impact applications using aramid fibers. I recommend including and discussing the following reference dealing with the topic:
https://doi.org/10.1016/j.dt.2021.09.009
The authors should find 3-5 more relevant references and discuss them.
Section 3
-It is clear from the results that the natural fiber is not improving the performance of the composite compared to the aramid fiber composite. This is also shown in the tensile test results reported by the authors in a previous publication. Delamination seems to be a big issue in the hybrid composites, which may be related to poor manufacturing and consolidation of the laminate. Since the paper's primary focus is an application for helmets against impact, the authors should add a discussion section at the end of section 3 highlighting that the material is not suitable for ballistic applications and discuss other possible applications for this hybrid composite.
Author Response
Manuscript Polymers-1856753
Response to Reviewers
The authors would like to thank the Reviewers for the valuable comments and suggestions that contribute to improve our manuscript. Amendments were provided accordingly and all modifications were marked as Track Changes in the revised version of the manuscript. Responses to each comment, point by point, are given below.
Reviewer #1 comments:
General comment: This manuscript presents the flexural characterization of hybrid laminated composites made of aramid and curaua natural fibers and epoxy resin. The composites are intended to be used as an armor material for helmet applications. The manuscript reports the methodology to test the composite experimentally. In general, the results are clear, well discussed, and of interest to readers of Polymers; however, the manuscript needs improvement. Therefore, the following comments should be addressed before the manuscript can be recommended for publication:
- The manuscript has several grammar, style, and syntax errors. The text is hard to read in the abstract and introduction. The manuscript should be checked thoroughly for grammatical mistakes and proofread by a native English speaker before it can be recommended for publication.
Response:
The authors are grateful for the considerations made by the reviewer, and as suggested, introduction and abstract were reviewed and approved by an expert in English language.
Introduction: The authors should add a few more references regarding hybrid composites for impact applications using aramid fibers. I recommend including and discussing the following reference dealing with the topic:
https://doi.org/10.1016/j.dt.2021.09.009
The authors should find 3-5 more relevant references and discuss them.
Response:
Excellent suggestion. As suggested, the indicated reference and also others that were considered important for the work were included. The authors initially did extensive an research on the topic and publications in the area. As can be seen below. However, it was removed from the introduction due to the length of the article.
“Several researches on hybrid composites based on synthetic fibers has been carried out. In Figure, more than 6,000 papers have been published using the keywords hybrid or hybridization epoxy-composite, according to the Scopus platform, including publications in 2023. However, only 921 works have explored natural fiber-epoxy hybrid composites. Considering hybridization with aramid, only 285 publications have been found to date. No publications considering curaua-aramid-epoxy hybrid compounds are found. As such, the novelty and importance of this study is evidenced.”
Section 3: It is clear from the results that the natural fiber is not improving the performance of the composite compared to the aramid fiber composite. This is also shown in the tensile test results reported by the authors in a previous publication. Delamination seems to be a big issue in the hybrid composites, which may be related to poor manufacturing and consolidation of the laminate. Since the paper's primary focus is an application for helmets against impact, the authors should add a discussion section at the end of section 3 highlighting that the material is not suitable for ballistic applications and discuss other possible applications for this hybrid composite.
Response:
The authors agree with the observations made by the reviewer, and as suggested, a textual excerpt was inserted highlighting other possibilities for using the material.

Reviewer 2 Report
1. Replace term Ballistic helmet with "impact applications" in title Curaua-Aramid Hybrid Laminated Composites for Ballistic Helmet: Flexural, Charpy Impact and Elastic Properties.
2. Rewrite conclusion and make it concise.
Author Response
Manuscript Polymers-1856753
Response to Reviewers
The authors would like to thank the Reviewers for the valuable comments and suggestions that contribute to improve our manuscript. Amendments were provided accordingly and all modifications were marked as Track Changes in the revised version of the manuscript. Responses to each comment, point by point, are given below.
Reviewer #2 comments:
Replace term Ballistic helmet with "impact applications" in title Curaua-Aramid Hybrid Laminated Composites for Ballistic Helmet: Flexural, Charpy Impact and Elastic Properties.
Response:
The authors are grateful for the observations made by the reviewer. Indeed, the reviewer's suggestion is a good one and will be taken into account regarding the title change.
Rewrite conclusion and make it concise.
Response:
As requested by the reviewer, the change in the conclusion was complied with and concisely.

Reviewer 3 Report
1. The abstract be modified and rewritten with the positive view point. For example, the authors be mentioned that hybridizing the Curaua with the Kevlar fiber caused to improve the mechanical properties than the Curaua/epoxy composite.
2. It is recommended that the Curaua in the first line of abstract be introduced “as leaves based natural fibers”. In other words, this phrase be used instead of “an Amazon natural fiber”.
3. In the introduction section (line 38), the percentage of cellulose, hemi-cellulose lignin and etc. of Curaua fibers be added. Also, the fibril angle of these fibers be mentioned.
4. In line 86, the weight fraction of epoxy to TETA is 100 to 13. It has been written in reverse.
5. The hybrid configuration be introduced in form of schematic image. It is unclear.
6. What is the surface modification method of Curaua fibers for using in the epoxy composite? Be added.
7. How many samples were examined for each mechanical test? Be added into the manuscript.
Author Response
Manuscript Polymers-1856753
Response to Reviewers
The authors would like to thank the Reviewers for the valuable comments and suggestions that contribute to improve our manuscript. Amendments were provided accordingly and all modifications were marked as Track Changes in the revised version of the manuscript. Responses to each comment, point by point, are given below.
Reviewer #3 comments:
The abstract be modified and rewritten with the positive view point. For example, the authors be mentioned that hybridizing the Curaua with the Kevlar fiber caused to improve the mechanical properties than the Curaua/epoxy composite.
Response:
The authors are grateful for the suggestions and, as recommended, the abstract was rewritten, highlighting the positive points of the material.
It is recommended that the Curaua in the first line of abstract be introduced “as leaves based natural fibers”. In other words, this phrase be used instead of “an Amazon natural fiber”.
Response:
Compiled, as recommended, once the suggestion is perfectly suited to the text.
In the introduction section (line 38), the percentage of cellulose, hemi-cellulose lignin and etc. of Curaua fibers be added. Also, the fibril angle of these fibers be mentioned.
Response:
As requested, information on the percentages of cellulose, hemicellulose and lignin was compiled from the literature and added to the revised version, as well as the microfibrillar angle of curaua fiber.
In line 86, the weight fraction of epoxy to TETA is 100 to 13. It has been written in reverse.
Response:
Well observed by the reviewer, the correction of the resin/hardener fraction was performed as suggested by the reviewer.
The hybrid configuration be introduced in form of schematic image. It is unclear.
Response:
The configuration was modified and, therefore, better reported in Figure 1, as suggested.
What is the surface modification method of Curaua fibers for using in the epoxy composite? Be added.
Response:
No method of surface modification was considered for cost-effectiveness the composite manufacture, that is, the curaua mat was used as commercially received.
How many samples were examined for each mechanical test? Be added into the manuscript.
Response:
For the Charpy impact and flexural test, eight and ten samples were considered of each proposed configuration (Table 1), respectively. Due to the non-destructive nature of the impulse excitation technique, one sample of each composite was used, and ten tests were performed per sample/composite.

Round 2
Reviewer 1 Report
The authors have addressed the reviewer's comments.
Author Response
Manuscript Polymers-1856753
Response to Reviewers
The authors would like to thank the Reviewers for the valuable comments and suggestions that contribute to improve our manuscript. Amendments were provided accordingly and all modifications were marked as Track Changes in the revised version of the manuscript. Responses to each comment, point by point, are given below.
Reviewer #1 comments:
General comment:
Open Review
(x) I would not like to sign my review report
( ) I would like to sign my review report
English language and style
( ) Extensive editing of English language and style required
( ) Moderate English changes required
(x) English language and style are fine/minor spell check required
( ) I don't feel qualified to judge about the English language and style
Yes Can be improved Must be improved Not applicable
Does the introduction provide sufficient background and include all relevant references?
( ) (x) ( ) ( )
Are all the cited references relevant to the research?
( ) (x) ( ) ( )
Is the research design appropriate?
( ) (x) ( ) ( )
Are the methods adequately described?
( ) (x) ( ) ( )
Are the results clearly presented?
( ) (x) ( ) ( )
Are the conclusions supported by the results?
( ) (x) ( ) ( )
Comments and Suggestions for Authors
The authors have addressed the reviewer's comments.
Response:
The authors once again thank the Reviewer for his zeal for English, with this we affirm that the text was fully revised by an expert in English.
